# Effect of a Health Education Intervention to Reduce Fear of Falling and Falls in Older People: A Cluster Randomized Clinical Trial Protocol

**DOI:** 10.3390/healthcare12242510

**Published:** 2024-12-11

**Authors:** Nuria Alcolea-Ruiz, Candelas López-López, Teresa Pérez-Pérez, Sonia Alcolea, Francisco Javier Pérez-Rivas

**Affiliations:** 1Sector III Healthcare Centre, South Assistance Directorate, Primary Care Assistance Management, Madrid Health Service, 28905 Madrid, Spain; 2Faculty of Nursing, Physiotherapy, and Podiatry, Complutense University of Madrid, 28040 Madrid, Spain; 3Emergency and Trauma Intensive Care Unit, 12 de Octubre University Hospital, 28041 Madrid, Spain; 4Invecuid Care Research Group, Hospital 12 de Octubre Health Research Institute (imas12), 28041 Madrid, Spain; 5“Salud Pública-Estilos de Vida, Metodología Enfermera y Cuidados en el Entorno Comunitario” Research Group, Nursing Department, Faculty of Nursing, Physiotherapy, and Podiatry, Complutense University of Madrid, 28040 Madrid, Spain; 6Department of Statistics and Data Science, Faculty of Statistical Studies, Complutense University of Madrid, 28040 Madrid, Spain; 7La Paz Children’s Hospital, La Paz Biomedical Research Foundation (IdiPAZ), CYBER INFECT, Madrid Health Service, 28046 Madrid, Spain; 8Primary Care Assistance Management, Madrid Health Service, 28035 Madrid, Spain; 9RICAPPS Research Networking Centre in Chronicity, Primary Care, and Health Promotion (RICORS), Carlos III Institute of Health, 28220 Madrid, Spain

**Keywords:** nursing, primary care nursing, primary health care, aged, frail elderly, fear, accidental falls, nurses, community health, cognitive behavioral therapy, exercise therapy

## Abstract

**Background/Objectives:** Fear of falling (FOF) and falls are prevalent issues among older adults, leading to activity restriction, decreased quality of life, and increased dependency. This study aims to assess the effectiveness of a nurse-led health education intervention to reduce FOF and fall incidence in older adults within primary care settings. **Methods:** This two-arm, multicenter, parallel, cluster-randomized clinical trial includes ten primary care centers in Spain and will enroll 150 adults over 65 years with FOF, mild or no functional dependence, and independent ambulation. Participants will be randomized to either the intervention group, which will receive five initial group education sessions led by community nurses and a booster session at six months, or the control group, which will receive usual care. Primary outcomes include FOF, assessed using the Short Falls Efficacy Scale-International (FES-I), and fall incidence. Data collection will occur at baseline, one month, six months, and twelve months post intervention. This study has been approved by the Ethics Committee for Research with Medicinal Products at Gregorio Marañón University Hospital in accordance with the Declaration of Helsinki. **Expected Outcomes:** The health education intervention is expected to significantly reduce both FOF and fall incidence, supporting the integration of FOF management in routine primary care for older adults, with potential benefits for safety and quality of life. Trial Registration: ClinicalTrials.gov: NCT05889910. The study protocol follows CONSORT and SPIRIT guidelines.

## 1. Introduction

Fear of falling (FOF) is a persistent concern about falling that causes individuals to avoid activities that they are still capable of doing [1]. In the long term, this problem increases the risk of falls [2], impairs functional ability in older people, [3,4] and even increases the risk of frailty, dependence, and death, affecting those most concerned about falls physically, socially, and psychologically [5,6,7].

FOF is common among community-dwelling older people, with its prevalence ranging from 26.9% to 49.4% worldwide [3,8,9]. Risk factors for FOF are manifold, including old age, being female, living alone, experiencing depression or anxiety, prior falls (especially those resulting in injuries, fractures and/or hospitalization), gait and balance disorders, chronic pain, pharmacological treatments (anxiolytics and antidepressants), and sensory deficits [10,11,12].

Moreover, according to the World Health Organization’s Global Burden of Disease report, fall-related injuries are the third leading cause of years lived with disability [13]. The prevalence of falls in people over 65 years of age is high. Approximately 30% have fallen once and more than 15% have fallen at least twice [14]. The annual incidence of falls ranges between 30% and 40% in those over 65 years of age and up to 50% in those over 80 years of age [15].

Risk factors associated with falls are also multiple, including being female, advanced age (over 80 years), unsuitable environmental factors, decreased ability to perform basic or instrumental activities of daily living, decreased functional ability (as an indicator of frailty), sensory deficit, comorbidities, polypharmacy, urinary incontinence, depression, low weight, and low income [2,10].

Traditionally, FOF has been considered as a ‘post fall syndrome’; however, studies have shown that, although concern about falling is more common in those who have experienced a fall, it can occur in individuals who have never fallen and should therefore be considered as an independent risk factor in the development of disability in older people [3].

Evidence on the management of FOF indicates that there are effective interventions to reduce FOF and overcome activity restrictions that can lead to disability in the field of primary care. The interventions that have been shown to be most effective are based on cognitive behavioral therapy programs combining multicomponent exercise, i.e., balance, gait speed, and lower limb strength training [16,17,18]. Other interventions that have been shown to be effective include tai chi [17], vitamin D supplementation, whole-body vibration therapy, and guided relaxation [19]. In addition, evidence on reducing the incidence of falls in community-dwelling older people suggests that fall prevention education programs and multicomponent exercise programs are the most effective interventions [20,21,22,23,24].

On the other hand, it is important to note that the use of nursing methodology to improve patient health outcomes is under development. According to Alfaro-Lefevre, the nursing care process is defined as a systematic and organized method of delivering personalized nursing care when responding to an actual or potential health disturbance, consistent with the basic approach of addressing the needs of each individual or group of individuals [25]. Several studies have identified that patients assigned to nurses who systematically and continuously use nursing methodology achieve better health outcomes [26,27]. The advantages of using this methodology include comprehensive patient care and reduced variability in the care provided by different professionals, which, in turn, facilitates the continuity of care [25]. Significant improvements have been documented in patients whose nurses use the nursing process in their routine clinical practice for intermediate outcomes, such as increased coverage in cervical cancer screening programs, increased vaccination coverage (rubella, tetanus, hepatitis B), glycemic control in patients with diabetes, perceived control of anxiety, and increased knowledge of chronic obstructive pulmonary disease [27,28,29,30]. Regarding the FOF and falls in older people in particular, according to Domínguez-Fernández’s doctoral thesis, published in 2021 [26], the regular use of nursing methodology (the application of care plans) improves the FES-I scale scores as well as the incidence of falls in people over 75 years of age compared to nurses who do not use it.

Primary care professionals play a key role in the early identification and treatment of FOF and falls in community-dwelling older people [8].

Therefore, given that (a) FOF and falls are interconnected, (b) they are highly prevalent problems, and (c) several studies show that it is possible to reduce concerns surrounding FOF and prevent falls by following intervention programs in people living in the community, we believe it is desirable to design and apply a standardized care plan that can be implemented in primary care facilities (PCFs) in a systematic way to address these problems in an ongoing and programmed manner, focusing on cognitive behavioral therapy, multicomponent exercise, and fall prevention.

## 2. Experimental Design

### 2.1. Aims

#### 2.1.1. Primary Objective

To assess the effectiveness of implementing a health education intervention to reduce FOF in community-dwelling individuals over 65 years of age.

#### 2.1.2. Secondary Objectives

To assess the effectiveness of implementing a health education intervention to reduce falls in community-dwelling individuals over 65 years of age.To analyze the relationships between patients’ sociodemographic, clinical, and functional variables and each of the following: the FES-I short questionnaire scores and the incidence of falls in both the control and intervention groups.To analyze the effect of a health education intervention on functional variables, emotional state, ability to practice self-care, and perceived health.

### 2.2. Research Hypothesis

Implementation of a standardized nursing care plan for at least six weeks reduces FOF (defined as patients’ confidence in being able to perform activities of daily living without losing balance or falling, as measured by the short FES-I scale) by at least 3.8 points among independent community-dwelling individuals over 65 years of age at six and 12 months post intervention compared to usual clinical nursing practice in these patients.

### 2.3. Design

The CONSORT (Consolidated Standards of Reporting Trials) and SPIRIT (Standard Protocol Items: Recommendations for Interventional Trials) guidelines were used to design this study. This is a parallel group, multicenter, open, two-arm, superiority cluster-randomized clinical trial with a 1:1 allocation ratio.

The PCFs participating in this study are randomly assigned to either the intervention or control arm. Participants randomized to the intervention group will receive a group-based health educational intervention based on the proposed protocol. Participants in the control group PCFs will receive care as usual.

### 2.4. Setting of This Study

This study will be carried out at ten PCFs in Madrid, Spain.

### 2.5. Recruitment of Facilities and Professionals

For recruitment purposes, researchers will contact PCFs and clinicians via an email sent by a scientific society of primary care nurses. Professionals interested in participating will be interviewed by the principal investigator.

Participating facilities shall have at least three nurse collaborators who will be able to deliver the proposed health educational intervention, as well as a suitable room available in the same basic healthcare district in which to implement the intervention in case they are assigned to the intervention group.

Participating professionals will become members of the FEARFALL_CARE Clinical Care Group. Each nurse will be recruiting approximately five participants. If a PCF is assigned to the control group, professionals will perform recruitment, study assessment interviews, and clinical practice as usual. If a PCF is assigned to the intervention group, at least three nurses shall participate in the recruitment and assessment interviews and at least two nurses will implement the educational group intervention (with one delivering the session and the other acting as an observer).

Finally, to minimize the risk of selection bias, PCF professionals shall recruit participants without the knowledge of the group to which they will be assigned.

### 2.6. Study Population

The study population will be individuals over 65 years of age who meet the eligibility criteria:

#### 2.6.1. Inclusion Criteria

Individuals over 65 years of age on the start date of this study.Individuals who are independent for activities of daily living or with mild functional dependence (Barthel Index score ≥ 60 and Short Physical Performance Battery (SPPB) ≥ 4).Independent for ambulation (can walk 45 m unaided or with a cane).Without cognitive impairment (Mini-Mental State Examination (MMSE) ≥ 24).With FOF (short FES-I ≥ 11).

#### 2.6.2. Exclusion Criteria

People with the following medical diagnoses or health conditions (coded according to the International Classification of Diseases (ICD–10):-Diagnosis of mental, behavioral, or neurodevelopmental disorders: delirium, dementia, amnestic disorders, or other cognitive disorders (F05.0; F05.9; F00; F02.8; F03; F04; R41.3; F06.9). Mental disorders caused by a general medical condition, not elsewhere classified (F06.1; F07.0; F09). Schizophrenia or other psychotic disorders (F20; F22; F23; F24; F29).-Diagnosis of neurodegenerative diseases: Parkinson’s disease (G20); Alzheimer’s disease (G30); multiple sclerosis (G35); myasthenia gravis, or other myoneuronal disorders (G70).-Diagnosis of blindness or low vision (H54).-Diagnosis of conductive or sensorineural hearing loss, bilateral or uncorrected with a hearing aid (H90.0; H90.2; H90.5; H90.6; H90.8) or other types of hearing loss (H83.3; H91), as long as it impairs participants’ understanding.-Diagnosis of acute ischemic heart diseases and cerebrovascular diseases in the previous year (I20–I24; I60–I63; I67; I68).-Other diseases of the circulatory system that contraindicate the multicomponent physical exercise program [31]: other forms of heart disease (I30–I52) (e.g., uncontrolled atrial or ventricular arrhythmias, dissecting aortic aneurysm, severe aortic stenosis, acute endocarditis/pericarditis, uncontrolled arterial hypertension, acute thromboembolic disease, severe acute heart failure, severe acute respiratory failure, uncontrolled orthostatic hypotension; diabetes mellitus with acute decompensation or uncontrolled hypoglycemia (E10-E14), recent fracture in the last month (T14.2) (for strength training) or any other circumstance that the professional considers prevents this type of patients from performing physical activity in the program.Hospitalization during the recruitment period or expected admissions during the study period.Institutionalized patients or with frequent changes in address.

#### 2.6.3. Withdrawal Criteria

Any participant who, after the beginning of this study, experiences events that may cause them to meet any of the exclusion criteria or to be in breach of the inclusion criteria shall be withdrawn from the trial. The data of the terminated subject will be collected until such time as non-compliance with the eligibility criteria is identified. No patients will be replaced, as the established sample size includes an additional 20% of participants to accommodate for potential losses.

### 2.7. Sample Size

We expect to be able to recruit ten PCFs (five assigned to the intervention group and five to the control group). Assuming a significance level of 0.05, a difference of 3.8 points on the Short Falls Efficacy Scale-International (FES-I), a standard deviation of 5.1 points, and an intraclass correlation coefficient of 0.1, at least 12 participants will be required per center (a total of 120 participants) to reach a power of 80%. Allowing for a loss of 20%, the necessary sample size will be 150 individuals (75 patients assigned to the intervention group and 75 to the control group; approximately 15 patients in each PCF, about five patients per nurse).

Patients seen by nurses at each participating facility and meeting the eligibility criteria will be recruited using a consecutive sampling method until the required sample size—which will be the same for all participating facilities—has been reached.

### 2.8. Randomization

Ten PCFs will be randomly assigned to the intervention or control group, resulting in five facilities per arm. An allocation ratio of 1:1 will be used. The randomization will be performed using a random number generator. The study coordinator will run the generator and place the allocations into sealed envelopes.

### 2.9. Study Blinding

The intervention can only be blinded during participant recruitment, when the professionals in the Clinical Care Group will not know to which group they will be assigned. To avoid interpretation bias throughout the statistical analysis, patient information will be encoded. The data analysis will be performed by a third party not involved in the fieldwork. The principal investigator will assess the results and participate in the fieldwork at one PCF, which means that she cannot be blinded when assessing the results obtained.

### 2.10. Training

To increase compliance with the protocols in place, the healthcare professionals in the Clinical Care Group will undergo accredited training, which will cover the information related to assessment and data collection in both study groups, as well as the implementation of the health education intervention in the intervention group. The professionals assigned to the intervention group will be trained separately in order to avoid compromising the indications given to the other arm of this study.

### 2.11. Study Variables

For further information, see Appendix A. Description of assessment tools.

#### 2.11.1. Primary Outcome Variable

The seven-item short FES-I scale (Spanish adaptation) scores will be collected at baseline, one month, six months, and one year after the health education intervention. The scale is validated for use with Spanish speakers, with a range of 7–28. A score equal to or greater than 11 is considered positive for FOF (see Table 1). The short FES-I scale showed a Cronbach’s alpha of 0.88 compared to the long version. In addition, the short FES-I obtained a Spearman’s rho correlation coefficient of 0.99 between T1 (measurement at baseline) and T2 (measurement at one month), as well as an intraclass correlation coefficient of 0.77 between T1 and T2 [32].

#### 2.11.2. Secondary Outcome Variable

Data on the incidence of falls at one month, six months, and one year after the health education intervention will be collected.

#### 2.11.3. Explanatory Variables

Sociodemographic variables: sex, age, marital status, number of cohabiting individuals, level of education, personal net monthly income, current or previous occupation, availability and type of support at home, perceived ability to receive family support, neighborhood, basic healthcare district, the Medea index (used to measure social inequalities in Spain), time spent in the neighborhood in years, housing characteristics, most frequent type of housing in the neighborhood, accessibility to basic services, perceived safety and walkability of the neighborhood.Functional variables: the Downton scale, the Barthel index, the SPPB scale total score, four-meter gait disturbance, balance disturbance, loss of lower limb strength, the Lawton-Brody index, physical activity recommended for this age group by the Spanish Ministry of Health (i.e., 150 min of moderate physical activity or 75 min of vigorous physical activity; balance exercises; muscle strengthening exercises; flexibility exercises). Data on this variable will be collected separately, and patients will be considered as meeting the recommended physical activity level, provided that 75% of the activities are carried out. This will include a variable regarding how many dimensions (from one to four) are met by the patient.Clinical variables: falls in the previous year (pre intervention); falls during intervention; fall-associated injuries (pre intervention, during intervention, and post intervention); mild visual problems or use of glasses; mild hypoacusis, either unilateral or corrected with a hearing aid; use of a walking device; pain (using a verbal numerical scale); anxiety (via the Generalized Anxiety Disorder-7 (GAD-7) questionnaire); depression (via the Patient Health Questionnaire-8 (PHQ-8); ability to practice self-care (via the Appraisal of Self-Care Agency Scale-Revised (ASA-R)); cognitive level (using the Mini-Mental State Examination); perceived health status; body mass index; urinary incontinence, comorbidities (using the Charlson index); medication; polypharmacy; high blood pressure; nursing diagnosis of risk of falling.Usual footwear variables: heel height in centimeters, type of sole, sole material, type of fastening.Feasibility and acceptability variables:○Feasibility (to be measured among professionals): recruitment rate, randomization, adherence to the health education intervention, follow-ups performed, safety/adverse events, difficulties in implementing the health education intervention. To be assessed at the end of the clinical trial.○Acceptability (to be assessed in patients assigned to the intervention group):-Participants: at the end of the health education intervention, data on its utility, usability, satisfactoriness, recommendability, as well as other open questions will be collected.-Professionals: data on the utility, usability, and satisfactoriness of the program, as well as their readiness to repeat it and other open questions will be collected.

## 3. Materials and Equipment

To carry out the health education intervention, at least two trained professionals per center are required, as well as a room with sufficient space and materials for participants to carry out the multicomponent exercise program explained in Appendix A. Educational program for nurses in the intervention group.

## 4. Detailed Procedure

This study has two arms, an intervention and a control arm. Participants from PCFs assigned to the intervention arm would receive the group educational intervention, while participants from PCFs in the control group would receive care as usual. Participants will be assessed for eligibility based on the inclusion and exclusion criteria at the nurses’ office. In cases where the patient agrees to participate and meets the criteria, a screening visit will be conducted to confirm compliance. Screening sheets will be kept for those participants who do not meet the criteria or do not wish to participate.

Patients will be followed up for 12 months. Data collection visits will take place at baseline, one month, six months, and 12 months.

### 4.1. Intervention Group Arm

The health education intervention will consist of five initial group sessions, plus a reinforcement session after six months. The sessions will be two hours long and will preferably be held once per week. Each group will have approximately 15 participants. The contents detailed in Appendix A. Educational program for nurses in the intervention group will be addressed.

The sessions will include nursing interventions aimed at improving participants’ knowledge, skills, and attitudes regarding their functional ability, fall prevention measures, anxiety management, and ability to cope with FOF and falls.

The activities to be performed during the intervention were drawn from standardized nursing interventions listed in the Nursing Interventions Classification (NIC) following a review of effective interventions addressing these issues [3,4,9,18,33]:NIC 1665. Functional ability enhancement.NIC 5612. Teaching: prescribed exercise.NIC 6490. Fall prevention.NIC 6486. Environmental management: safety.NIC 4700. Cognitive restructuring.NIC 5820. Anxiety reduction.NIC 5230. Coping enhancement.

To monitor the degree of compliance with the interventions as instructed in the protocol, ad hoc questionnaires will be sent after each session to all the professionals and patients who carry out the intervention in order to assess acceptability, feasibility, and satisfaction.

Health education workshops are a fundamental tool for teaching health education in the primary care setting. This workshop was designed following the methodology recommended by the Madrid Primary Care Management Board and is being reviewed by the Validation Commission for Educational Projects of the Community of Madrid (COVAM) for its free distribution throughout the Autonomous Community of Madrid.

### 4.2. Control Group Arm

Each patient in the control group will receive the usual clinical care from their respective nurse in the primary care setting for the duration of this study. The usual clinical practice involves addressing users’ requirements and delivering interventions that can have a positive impact on FOF and/or falls. These activities/interventions are based on the Portfolio of Standardized Services of the Autonomous Community of Madrid, which includes these patients due to their age or as a consequence of their chronic conditions, while detailing the steps to be taken by healthcare professionals.

The nurse will provide structured frailty reversal education in a six-monthly action plan to those patients assigned to the control group, in whom the nurse detects risk of frailty-induced falls (SPPB score below 10, Barthel score between 91 and 100). This action plan will be the one proposed in the Portfolio of Services of the Primary Care Management Board: nutrition, moderate sun exposure, multicomponent physical exercise, fall prevention, and maintenance of intellectual activity and social relations, as well as review of the prescribed treatment, appropriate use of medication, and adherence to treatment. In the case of moderately dependent patients (Barthel score between 61 and 90), the nurse will provide them with a six-monthly action plan covering the proposed structured education: nutrition, physical exercise, fall prevention, and use of mobility aids, as well as review of the prescribed treatment, appropriate use of medication, and adherence to pharmacological treatment.

Figure 1 shows the CONSORT flowchart of participants. Patients participating in this study will not be able to participate in any other study involving educational interventions while this study is ongoing. Patients participating in this study will only be with-drawn from it should they meet withdrawal criteria or no longer wish to participate. No specific treatments have been considered that would preclude continuation in this study.

### 4.3. Visit Plan

During the study period, both study groups will make the following visits (see Table 2):-Initial visit: screening.-Pre-intervention baseline visit: Collection of data on variables. Approximate duration: 60–80 min. The patients assigned to the control group shall carry it out up to two weeks before the beginning of the initial health education intervention.-Five initial sessions with the intervention group.-Visit post initial intervention (visit window: ±4–8 weeks after randomization (for the control group) or ±2 weeks after the initial sessions (for the intervention group)): Collection of data on variables. Approximate duration: 25 min.-Visit at six months post initial intervention (visit window: ±2–4 weeks, at six months after randomization (for the control group) or after the initial health education intervention (for the intervention group)), always prior to the booster session. Collection of data on variables. The structured education plan proposed in the Portfolio of Services will be reviewed in both groups, as well as in the patients in the intervention group, including a follow-up of the issues identified during the health education intervention. Approximate duration: 45 min.-Booster session with the intervention group.-Visit at 12 months (visit window: ±2–4 weeks, at 12 months post randomization in the control group or at 6 months after the booster session in the intervention group). Collection of data on variables. Approximate duration: 20 min.

### 4.4. Data Analysis

A descriptive analysis by groups (intervention and control) of participants’ sociodemographic, functional, and clinical characteristics will be carried out to identify potential differences between the two study groups, with quantitative variables expressed as either means and standard deviations or medians and interquartile ranges depending on the normality or skewness of data distribution. Qualitative variables will be summarized as frequencies and percentages. Data on the effect of the health education intervention will be analyzed in both the intention-to-treat population and the per-protocol population. In the former case, all participants will be included and will be analyzed as part of the group to which their PCF will be randomly assigned. Should the percentage of missing data be higher than 5%, multiple imputation techniques will be used. In the per-protocol analysis, complete cases of patients will be included provided they have maintained adherence. The intervention group is required to be adherent and have at least 80% attendance at the initial five sessions. Safety, acceptability, and feasibility of the health education intervention will also be assessed.

The primary efficacy analysis, (FOF), will be the comparison between intervention and control groups fitting a fixed-effects ANOVA model. Model will include group and facility as independent variables, with facility being a nested factor within group. The short FES-I scale mean scores and their 95% confidence intervals will be provided for each facility, or by groups if no differences between facilities are observed. The effectiveness of the health education intervention in relation to fear of falling will be measured through the decrease in the short FES-I scale scores, assessed at three times: at 1-, 6- and 12 months after the intervention. The difference in means between the intervention and control groups will be analyzed. In addition, the variable will be examined dichotomously, using a cut-off point on the short FES-I scale of 11 or more to identify the presence of fear of falling. Multivariate models will be fitted to identify factors potentially related to FOF.

For the secondary outcome variable (incidence of falls), a process similar to the main analysis will be followed, but this time fitting a fixed-effects logistic regression model, again nesting each facility within its group. The effectiveness of the health education intervention for the incidence of falls will be measured through the recording of falls (recorded in the medical history or reported by the patient), at 1-, 6- and 12-months post intervention. Both groups will be compared in terms of odds ratios, and their corresponding 95% confidence intervals will be provided.

To assess the impact of the health education intervention by time, mixed effects models will be adjusted, including time also as an independent variable and random intercepts for participants.

Data will be recorded in a paper data collection notebook and entered into an Excel spreadsheet by a single researcher in order to minimize the risk of transcription errors. The collected data will then undergo a second review to correct transcription errors, if necessary. Access to these data will be limited to authors until open access dissemination, after which they can be included on ClinicalTrials.gov.

### 4.5. Validity and Reliability/Rigor

To ensure adherence to the protocol and that the data are collected in the same way by the different professionals, it will be compulsory to take an accredited training course with theoretical and practical content in order to join the Clinical Care Group, with all information being provided in writing. In addition, the facilities in the intervention group will receive specific and accredited training on the health education intervention.

Additionally, the NIC taxonomy [34] will be used to standardize the interventions that will be implemented and unify the steps to be taken by professionals. Furthermore, recruitment will be blinded for the participating professionals to avoid selection bias in the different groups, and randomization will take place once the informed consent of all participants has been obtained.

It is also important to note the rationale for using a cluster or parallel cluster randomized clinical trial (RCT) design. Cluster RCTs are considered the most suitable designs for validating interventions that will be implemented at the organizational rather than individual level [35]. Moreover, the implementation of clusters for each facility would avoid potential “contamination” between patients receiving the health education intervention and those in the control group by reducing the dilution bias of the intervention effect [36] and improving the rigor and consistency of each of the study groups.

A number of strategies have been used in the study design to avoid or reduce potential biases associated with the cluster design such as the possibility of unbalanced groups in terms of magnitude, unequal attrition rates between groups, and selection bias in different groups. These strategies include ensuring that the groups are balanced in terms of: magnitude before the end of the recruitment period; recruiting patients, taking attrition rates into consideration in both groups; applying the same inclusion and exclusion criteria to both groups; and training research professionals to ensure that informed consent and data collection are obtained in the same way from both groups.

### 4.6. Clinical Risk Management

Clinical risk management has been considered to ensure the safety of participants in all phases of this study. Health education interventions to reduce fear of falling in older people at the community level have shown positive results in various studies [16,17,18,19,20,21,22,23,24]. For this study, those with evidence support have been selected, which is a guarantee of quality.

To ensure that no subjects with unfavorable conditions participate in this type of intervention at the community level and that no situations arise that create a risk to the health of the participants, measures will be taken at different stages of this study [37]:-Screening and pre-selection visits will allow the assessment of the baseline physical and mental condition of the participants, which will ensure whether they can safely participate in the multicomponent exercise program. Inclusion criteria include that participants must be independent for activities of daily living or with mild functional dependence, independent for walking, and not present cognitive impairment (details are given in the inclusion and exclusion criteria section).-The intervention visit will be carried out by nurses specifically trained in the field of primary care to implement the program, who will monitor the performance of the activities and ensure their adaptation to the functional capacities of each individual. Homogeneous training of the professionals involved in this study will ensure the safe implementation of the intervention.-Follow-up visits will allow close monitoring of each participant to monitor and detect possible conditions that pose a risk to health.-If during the intervention/follow-up a change in the patient’s clinical condition is detected that is a criterion for withdrawal, to ensure patient safety, the patient will not continue in this study. In addition, the patient will be followed up either in primary care or, if necessary, in specialized care.

In addition, this study has been approved by the Ethics Committee of the Gregorio Marañón Hospital and the Central Research Committee of the Primary Care Management of Madrid that guarantee the application of good research practices in the field of primary care. This study is considered a low-intervention clinical trial, in which the additional burden on the safety of the subjects is minimal compared to that of normal clinical practice and does not require coverage by an insurance contract [38].

Finally, this study is carried out on subjects covered by the National Health System, specifically in Spain, health coverage includes access to common services that guarantee adequate, comprehensive and continuous health care for all users, including public health services, primary care, specialized care, emergency care, pharmaceutical care, orthoprosthetic care, dietary products and medical transport [39].

Therefore, in the case of this type of study (a low-intervention clinical trial), immediate risk management will be covered and guaranteed by the National Health System healthcare for all participants.

## 5. Expected Results

The purpose of this study is to explore the possibility of improving FOF and the incidence of falls in community-dwelling people over 65 years of age through a group educational intervention carried out by nurses. This is the first study conducted in Spain, with the main objective of reducing FOF through nursing interventions in the primary care setting. This study will measure the effectiveness of the health education intervention at 1, 6 and 12 months, which means that the effect of the health education intervention will be assessed both in the short and long term.

This study protocol attempts to unify effective interventions into a single educational group intervention that could be implemented in a standardized way in the primary care setting. After a review of the relevant literature on effective interventions for this type of patient, we believe that a single intervention focusing on functional ability enhancement, fall prevention, cognitive restructuring, ability to cope, and anxiety management could represent the priority type of intervention for these patients, as reported in previous publications on the topic [3,4,9,18,19,21,22,24,35,40,41].

Moreover, the multicenter approach of this study and the data collected on these variables will help to assess the feasibility of incorporating the health education intervention into daily clinical practice.

If the health educational intervention proves effective, it could be of great value to patients living with this prevalent health problem, as it would facilitate the screening of FOF and the application of this type of intervention in nurses’ offices, thus helping to reduce long-term disability caused by voluntary activity restriction at earlier stages of life. Furthermore, the results of this study may contribute to future research aimed at addressing these issues among community-dwelling individuals over 65 years of age.

It is also our belief that the 12-month follow-up described above may yield valuable data for future studies in this line of research, as the duration of clinical trials in this line of research is usually no longer than 6 months [17,42,43]. In addition, it is important to know whether the effect of the group-based educational intervention has a long-term impact on patients, contribute providing long-term evidence on effective health education interventions to reduce fear of falling and falls in older people living in the community. Finally, this study could be added to other studies with results after 12 months, evaluating the sum of long-term research, also encouraging long-term research.

### Limitations

This study features a number of drawbacks. Firstly, a selection bias may be introduced during recruitment at PCFs. The recruitment of participants who are visiting their healthcare professionals could influence the results. However, it is not possible to recruit them randomly using databases, as there are no registers of patients with FOF. Secondly, given the nature of this study, the health education intervention cannot be blinded to participants. Thirdly, the validity of the data to be extracted from health records could be underestimated in cases with poor documentation. Finally, participants will be recruited only in the Autonomous Community of Madrid due to financial and human resources constraints, which may limit the generalizability of the results.

Difficulties may also be encountered when implementing this study, as it is an health education interventional trial with four scheduled visits, in which attrition rates may be high due to dropouts, non-attendance, or non-compliance with instructions. To mitigate this problem, we aim to maximize patient retention through regular telephone calls, in which appointments will be arranged with participants prior to each interview. Furthermore, an additional 20% of participants will be recruited to accommodate for potential losses.

## 6. Conclusions

Once this study is completed, it will be possible to assess the effectiveness of the proposed health education intervention in improving FOF and the incidence of falls in the older population.

A 12-month follow-up may provide valuable information on the sustained impact of such interventions, which could serve as a basis for future research, encouraging the adoption of longer evaluation periods.

If the health education intervention is shown to be successful, it shall be recommended that the entire procedure be widely integrated into routine practice in primary care settings.

## Figures and Tables

**Figure 1 healthcare-12-02510-f001:**
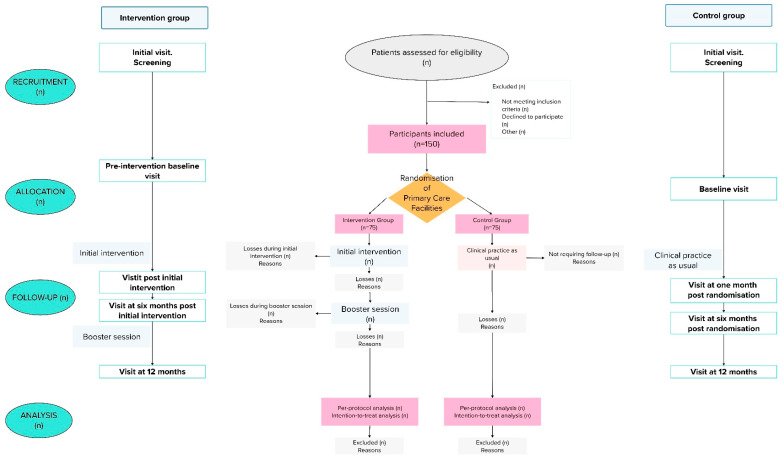
CONSORT flowchart of participants.

**Table 1 healthcare-12-02510-t001:** Short FES-I scale.

Short FES-I Scale How Concerned Are You About the Possibility of Falling While…?	Not at All Concerned	Somewhat Concerned	Fairly Concerned	Very Concerned
1. Getting dressed or undressed	1 □	2 □	3 □	4 □
2. Taking a bath or shower	1 □	2 □	3 □	4 □
3. Getting in or out of a chair	1 □	2 □	3 □	4 □
4. Going up or down stairs	1 □	2 □	3 □	4 □
5. Reaching for something above your head or on the ground	1 □	2 □	3 □	4 □
6. Walking up or down a slope	1 □	2 □	3 □	4 □
7. Going out to a social event (e.g., religious service, family gathering or club meeting)	1 □	2 □	3 □	4 □
SUBTOTAL:	Add all the 1’s	Add all the 2’s	Add all the 3’s	Add all the 4’s
	Short FES-I ≥ 11: Fear of falling			

**Table 2 healthcare-12-02510-t002:** Summary of visits and variables.

Variables	Visits						
	Initial Visit. Screening	Pre-Intervention Baseline visit ^1^	Five Initial Group Sessions	Visit Post Initial Intervention ^2^	Visit at Six Months Post Initial Intervention ^3^	Booster Session	Visit at 12 Months ^4^
Review of eligibility criteria	x						
Informed consent	x	x					
Primary outcome variable: Short FES-I		x		x	x		x
Secondary outcome variable: Post intervention falls		x		x	x		x
EXPLANATORY VARIABLES							
Sociodemographic variables		x					
Functional variables		x			x		
Usual footwear		x			x		
Clinical variables		x					
Falls during intervention				x		x	
Ability to practice self-care		x		x	x		x
Perceived health and anxiety		x		x	x		x
Acceptability (among patients)				x			

^1^ ±2 weeks before the health education intervention (for the intervention group: (IG)). ^2^ ±2 weeks after the initial sessions (IG). ±4–8 weeks after randomization (for the control group (CG)). ^3^ ±2–4 weeks, at six months after randomization (CG) or after the initial health education intervention (IG), always prior to the booster session. ^4^ ±2–4 weeks, at 12 months post randomization (CG), at six months after the booster session (IG).

## Data Availability

Data are contained within the article and Appendix A.

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
