# Peer review of "Effect of a Health Education Intervention to Reduce Fear of Falling and Falls in Older People: A Cluster Randomized Clinical Trial Protocol"

_healthcare, 2024, doi:10.3390/healthcare12242510_

Round 1
Reviewer 1 Report
Comments and Suggestions for Authors
Dear Authors,
The present research article, entitled “Effect of an Intervention to Reduce Fear of Falling and Falls in Older People: A Cluster Randomised Clinical Trial Protocol”, aims to assess the effectiveness of a nurse-led educational intervention to reduce FOF and fall incidence in older adults within primary care settings.
After reviewing the paper, it was determined that there was great rigor in establishing the randomization of the groups, the inclusion/exclusion criteria and the measures to avoid bias. In addition, the procedure to be followed and the expected results are explained in detail. I consider that the objective of the study is very well stated and the whole procedure is in accordance with it.
I believe that the article is suitable for publication.
I declare no conflict of interest regarding this manuscript.
Best regards.

Author Response
Dear reviewer,
Thank you very much for your response.
Sincerely,
Nuria Alcolea
Reviewer 2 Report
Comments and Suggestions for Authors
Dear Authors,
I had the pleasure of reading your manuscript, and I found it both engaging and well-written. The topic is highly relevant, especially considering the demographic shifts that highlight the importance of your study's theme. Below are several suggestions that I hope will help you refine the paper:
The introduction is well-constructed and nearly comprehensive in addressing the subject of falls. However, it is surprising that falls are not examined from a hospital perspective, particularly regarding hospital-based prevention. This is noteworthy given that one of the study's stated objectives is prevention, and both the hypothesis and study design reference nurses, by extension connecting to the hospital environment. Please consider including the following reference to support this angle: https://doi.org/10.1177/25160435241246344
There is a typographical error on line 111.
Could you clarify how you measured the effectiveness of a nurse-led educational intervention in reducing fear of falling (FOF) and the incidence of falls in older adults within primary care settings? A more detailed description of the measurement approach would strengthen the reliability of the study.
Why were patients with a diagnosis of acute ischemic heart disease excluded? Could you provide more specificity? Please indicate whether defined clinical criteria (e.g., ejection fraction, NYHA scale) underpinned this exclusion.
Was a statistician involved in conducting the statistical analyses?
Although the study uses nursing data, it does not adequately address clinical risk management, which would be an essential addition.
Finally, the lack of a conclusions section is noticeable. Adding one could improve the manuscript's structure by summarizing key findings and potential implications.
Thank you
Author Response
Dear Reviewer:
We would like to express our sincere gratitude for the positive evaluation of our study and for the detailed and constructive comments you have provided. We consider that your observations reflect a deep and enriching analysis, which will undoubtedly contribute to improving the quality and impact of our work.
We have carefully reviewed each of your suggestions and have made the necessary changes to comprehensively address the issues raised.

Reviewer 3 Report
Comments and Suggestions for Authors
- Please clearly and specifically describe the intervention in the title and objectives.
- I recommend unifying the terminology for "intervention" and "education" throughout the paper.
- Provide the reliability and validity of the variables.
- The paper mentions that the 12-month follow-up could be helpful for future research, but it would be beneficial to include a clear discussion of how it may contribute to future studies.
- In investigating the incidence of falls among study participants, what plans and compensations are in place for participants who may suffer fractures due to falls that require surgical intervention?
Author Response

(The authors gave the same response as above.)

Round 2
Reviewer 2 Report
Comments and Suggestions for Authors
Dear authors,
I hope my suggestions have been helpful.
Regarding the exclusion criteria, please integrate them within the paper by referring to the guidelines you shared. Otherwise, they are unclear. Regarding the development of the short EDF-I scale scores, please include them in a table. Paragraph 4.6 on clinical risk is short and confusing with those modalities. Therefore, I think you should address the focus of falls in the hospital and prevention strategies in clinical risk clearly and not in a confusing manner without any scientific citation.
Kind regards
Author Response
Dear Editor,
We appreciate the careful review and constructive comments. We believe that the revised manuscript is substantially improved after making the suggested edits.
With these modifications and improvements to the manuscript, we hope that it is now suitable for publication in Healthcare.
Kind regards.
